# The role of education, religiosity and development on support for violent practices among Muslims in thirty-five countries

**Aaron Gullickson** *, **Sarah Ahmed**

Department of Sociology, University of Oregon, Eugene, OR, United States of America

* aarong@uoregon.edu

## Abstract

Despite widespread scholarly interest in values and attitudes among Muslim populations, relatively little work has focused on specific attitudes popularly thought to indicate anti-modern or anti-liberal tendencies within Islam. In this article, we use data from the Pew Research Center from 2008-2012 to examine support for violent practices among Muslims in thirty-five countries. Support for violent practices is defined by three questions on the acceptability of killing apostates, the stoning of adulterers, and severe corporal punishment for thieves. Using multilevel models that capture country-level variability, we analyze the relationship between support for violent practices and education, religiosity, and development. In general, we find that support for violent practices is less common among individuals with more education and less religiosity and who come from more developed countries. However, when we examine variation across countries, we see evidence of substantial heterogeneity in the association of education and religiosity with support for violent practices. We find that education is more liberalizing in more liberal countries and in less developed countries. The effects of religiosity are also related to country-level context but vary depending on how religiosity is measured. Overall, the variation we observe across countries calls into question a civilizational approach to studying values among Muslim populations and points to a more detailed multiple modernities approach.

## Introduction

In a viral appearance on the HBO Show "Real Time with Bill Maher" in 2014, actor Ben Affleck became embroiled in an unexpected debate with host Bill Maher and fellow guest, Sam Harris, over the allegedly authoritarian and violent tendencies of Islamic doctrine [1]. Harris, one of the leading figures in the New Atheist movement, argued that Islam "at this moment is the motherlode of bad ideas." To support this view, Maher offered data from a recent poll of Muslims worldwide conducted by the Pew Research Center which showed that "like 90%" of Egyptian Muslims supported death as a punishment for leaving the faith.

Although Maher's figure was factually correct, the statistic is nonetheless misleading. Egyptian Muslims were more supportive of this statement than Muslims in any other country in

**Data Availability Statement:** All data and code uses for the analysis in this article are available on a GitHub repository at https://github.com/AaronGullickson/muslim_attitudes.

**Funding:** The authors received no specific funding for this work.

**Competing interests:** The authors have declared that no competing interests exist.

the Pew study. In contrast, only 0.8% of Muslims in Kazakhstan felt similarly. Among the more than thirty countries in the cited Pew survey as well as a similar one conducted in sub-Saharan Africa, the level of support among Muslims for killing apostates varies uniformly between these two extremes. Substantial variation rather than strict orthodoxy is the key to understanding Muslim responses to this question. For social scientists, this variation then begs a further question—why do Muslims vary so much in their support for such "bad" ideas?

In this article, we take up this question with a more formal analysis of the Pew data. The data were collected between 2008–12 on Muslims in thirty-five countries representing roughly sixty-five percent of the global Muslim population. From this data, we develop a value construct measuring support for violent practices from three questions on the acceptability of killing apostates, the stoning of adulterers, and severe corporal punishment for thieves. Following prior work on the impact of modernization and development on liberal values in mostly Western contexts, our analysis focuses on the relationship between this value construct, education, religiosity, and development.

Our analysis builds on an extensive literature on values and attitude among Muslims, and complements prior research in two important ways. First, we address specific attitudes that have never been examined in scholarly work, despite their prominence in popular discussions of Islam in the West. Most prior research focuses either on attitudes regarding universalistic and abstract concepts such as support for democracy, political violence, and gender equality, or relies upon large aggregate indices that are driven by a multitude of specific attitudes ranging from acceptance of abortion to whether respondents would be willing to sign a petition [2].

A focus on universalistic attitudes, however, provides little insight into the specific attitudes and values that differentiate cultural and religious groups. For example, whether a respondent is willing to sign a petition or is supportive of democracy in the abstract may tell us little to nothing about how that respondent feels toward the mandate that apostates should be killed. Yet, the justifiability of killing apostates is a topic that both divides Muslims internally and distinguishes Muslims from other religious groups. It is not simply that Muslims feel differently on this issue in comparison to other religious groups, but rather that the question itself is only intelligible among Muslims.

Second, the large sample of countries here allows us to more systematically examine country-level variation than has been possible in much of the prior work on value systems in Muslim populations. While proponents of classic modernization theory expected a fairly uniform pattern of modernization, contemporary scholars expect greater heterogeneity in this process [3]. Some have even suggested that change in cultural systems—including values and attitudes—may be largely unconnected to processes of modernization and development [4]. To better understand this heterogeneity, we use multilevel models that provide information about country-level variation in support for violent practices and country-level variation in the association between support for violent practices and education and religiosity. We argue that the pattern of variation at the country level provides insight into the role of educational and religious institutions in the process of value change.

In summary, our analysis is guided by the following research questions:

1. What is the relationship between support for violent practices and education, religiosity, and level of development across all countries?

2. How does the level of development in a country affect the relationship between support for violent practices and education/religiosity?

3. How is the average level of support for violence in a country associated with the relationship to education/religiosity in that country?

4. How are the country-specific effects of education and religiosity on support for violent practices associated with one another?

When we pool results across countries, our analysis somewhat supports the expectations of modernization theory. Support for violent practices is less common among individuals with more education and less religiosity and who come from more developed countries. However, when we examine variation across countries, we see evidence of substantial heterogeneity in the effects of education and religiosity on support for violent practices that is more consistent with a "multiple modernities" perspective. Further analysis of these patterns of heterogeneity suggest that the effects of both education and religiosity respond to the social context of the country in important, but also sometimes unexpected, ways.

## Background

Prior work on values and attitudes in Muslim populations has focused on a variety of such attitudes including support for democracy [5–7], political violence [8, 9], women's rights [10], and fundamentalism [11]. While a summary of this work is beyond the scope of this article, such studies are often framed around modernization theory and its closely related cousin, secularization theory, as well as critiques of these theories [6, 12, 13].

Modernization theory treats cultural and value change as largely a by-product of structural development [14–16]. Industrialization and economic growth bring about higher educational attainment, greater differentiation in the occupational workforce, urbanization, and higher standards of living [17]. All of these factors contribute to a shift from conservative and authoritarian values to liberal values that promote individual autonomy and self-determination [15]. Concomitantly, rationalization, individualism, and cultural pluralism reduce the significance of religious institutions in favor of secular institutions, diminishing the role of religion in social life [18–20].

From the basic expectation of modernization theory, we develop our first hypothesis:

*H1*: The level of development in a country is negatively associated with support for violent practices.

While classic modernization theorists have viewed modernization as a fundamental process that would evolve similarly across countries, many scholars today criticize this view for privileging the historical experience of the West as a universal model. These "multiple modernities" scholars argue that while modernity will likely lead to changes in social and cultural systems, these changes will materialize in multiple dissimilar and non-convergent forms because countries develop within various historical and cultural contexts [21–24].

In this article, we focus on the role that such heterogeneity may play in the association between support for violent practices and individual level variables across countries. Specifically, we look at the role of education and religiosity on value change, because these two variables are seen as key variables that connect individual-level practices to processes of modernization and value change [25]. The effects of these variables also inform us more broadly about the role of educational and religious institutions in value change.

### The role of education and religiosity in value change

Education is often seen as a crucial fulcrum at the individual level for processes of modernization [26–28]. In terms of value change, research has repeatedly shown that education is positively related to liberal values [29–32].

While these results are broadly consistent with modernization theory, significant debate has persisted on how to understand this finding. The development perspective, often tied to

modernization theory, holds that education is linked to fundamental cognitive changes in the way people view the world that lead to more tolerance and less authoritarianism [33, 34]. On the other hand, the socialization perspective argues that educational institutions socialize individuals to the prevailing norms of a society [35–37]. Therefore, the liberalizing effects of education will only exist in countries where liberal values are commonplace. Consistent with the socialization perspective, cross-national results in Western countries show that the positive effect of education is greatest in countries with more liberal values overall [38].

From these arguments, we develop several hypotheses relating to the effects of education. First, the development perspective would expect education to have a similarly negative effect on support for violent practices across countries. Thus:

*H2a*: The association between education and support for violent practices is consistently negative across countries.

A less stringent version of this hypothesis would expect that the liberalizing effects of education are most substantial in countries that have developed a modern educational system. Thus, we should expect the negative effect of education to be the greatest in countries with the highest level of development:

*H2b*: The association between education and support for violent practices is more negative in countries with higher levels of development.

Alternatively, the socialization perspective would expect that the effect of education will be the most negative in countries where support for violent practices is low and the negative effect of education will be weaker or even reverse direction in countries where support for violent practices is high. Thus:

*H2c*: The association between education and support for violent practices is positively correlated with overall support for violent practices in a country.

Finally, according to Huntington's civilizational argument, Western liberal traditions are not intrinsic to the process of modernity but rather a specific inheritance of Western civilization [4]. Thus, we should not expect to see education operate in a similar fashion outside of Western contexts. Specifically, Huntington argued that the primary agents for the contemporary rise of Islamic fundamentalism were an urban and educated middle class [4]. Rapid economic growth and urbanization dislocate individuals from traditional structures of social support and this dislocation is experienced most intensely by upwardly mobile, young, urban, and educated individuals. Therefore, contrary to research primarily in Western contexts showing a liberalizing effect of education, Huntington expected the reverse in Muslim societies. Thus:

*H2d*: The association between education and support for violent practices is consistently positive across countries.

Social scientists have often treated religiosity as the anti-modern counterweight to education [39]. High levels of religiosity are seen to indicate commitments to conservative and authoritarian values [40–42]. Although there is significant heterogeneity across countries, research has shown that religious identification tends to be associated with support for right-wing parties and more conservative attitudes [42–46].

Given this prior work, we develop a hypothesis that is analogous to H2a for education, but in the opposite direction:

*H3a*: The association between religiosity and support for violent practices is consistently positive across countries.

Similarly, we also consider a less stringent test which allows the level of development in a country to dampen this conservative effect of religiosity:

*H3b*: The association between religiosity and support for violent practices is less positive in countries with higher levels of development.

Although less widely acknowledged than for the case of education, the effect of religiosity may similarly be mediated by the role that religious institutions play as socialization agents [25]. In particular, religious institutions may respond to the prevailing norms and beliefs in a society in one of two ways. Greater liberalism within a society may dampen the conservative effect of religiosity on liberal values or it may alternatively provoke a greater defense of religious values and increase the association between religiosity and conservative values [47]. We therefore consider two hypotheses about the role that overall support for violent practices may play in the relationship between religiosity and that support.

*H3c*: The association between religiosity and support for violent practices is positively correlated with overall support for violent practices in a country (socialization).

*H3d*: The association between religiosity and support for violent practices is negatively correlated with overall support for violent practices in a country (resistance).

Importantly, the relationship between religiosity and conservative values may be partially tautological depending on how religiosity is defined. Given that the core religious texts of the major Western religions are frequently conservative and authoritarian, it is not surprising that strong believers with a tendency for theological literalism would themselves be more conservative and authoritarian. Therefore, in analyzing the effect of religiosity on values and attitudes, it is imperative to distinguish the effects of theological conservatism and literalism from the effects of variables that measure how important religion and religious practices are in a person's life.

Finally, we consider how the effects of education and religiosity within a country may be correlated to one another. This correlation tells us the degree to which unmeasured factors producing heterogeneity in these effects across countries operate similarly or differently on educational and religious institutions. If this correlation is positive, then we observe a process of synchronicity in which educational and religious institutions are responding in the same way to other societal influences. Alternatively, if the correlation is negative, then we observe a process of polarization between education and religious institutions, in which these societal influences push educational and religious institutions in opposite directions. Formally:

*H4a*: The effects of education and religiosity on support for violent practices are negatively correlated across countries (polarization).

*H4b*: The effects of education and religiosity on support for violent practices are negatively correlated across countries (synchronization).

## Analytical approach

To test the hypotheses outlined above, we developed a multilevel model using the Pew data detailed below. Specifically, we use a random intercept and random slope multilevel model in which the intercepts and the slopes for the educational attainment and religiosity variables are allowed to vary across countries. Multilevel models estimate country-specific parameters using partial pooling between the global estimate and fixed-effects country-specific estimates, which avoids overfitting the data and produces more reasonable estimates [48].

Our general model specification is:

$$y_{ij} = \beta_{0j} + \beta_{1j}(\texttt{religiosity}_{ij}) + \beta_{2j}(\texttt{education}_{ij}) + \sum_{k=1}^{K} \lambda_k x_{ijk} + \epsilon_{ij}$$

where $y_{ij}$ is the score on the measure of support for violent practices (detailed below) for the $i$th respondent in the $j$th country. The intercept $\beta_{0j}$, and the two slopes of $\beta_{1j}$ and $\beta_{2j}$ on religiosity and education, respectively, are allowed to vary by country. The model also includes $K$ additional independent control variables $x_{ijk}$ that are not allowed to vary by country. $\epsilon_{ij}$ is an individual-level error term.

To complete the multilevel model structure, the three country-varying parameters have their own linear equations:

$$\beta_{0j} = \gamma_{01} + \gamma_{02}(\texttt{development}_j) + \delta_j$$

$$\beta_{1j} = \gamma_{11} + \gamma_{12}(\texttt{development}_j) + \eta_j$$

$$\beta_{2j} = \gamma_{21} + \gamma_{22}(\texttt{development}_j) + \tau_j$$

The structure shown here is for the fullest model in which we include a country-level measure of development (detailed below) that predicts the intercept and the two slopes. The $\gamma_{02}$ term measures the association between development and average support for violent practices in a country, which directly addresses *H1*. The $\gamma_{12}$ and $\gamma_{22}$ terms measure how development in a country affects the individual-level relationship between support for violent practices and either education or religiosity, respectively. These terms allows us to test *H2b* and *H3b*.

The error terms $\delta_j$, $\eta_j$ and $\tau_j$ are country-level random effects that measure country-level heterogeneity in terms of the intercept and two slopes. The country-specific intercepts provide a measure of the overall level of support for violent practices within a given country, while the country-specific slopes provide of the association of education and religiosity to support for violent practices within a given country. The variance of these random effects and the correlation between them allows us to address the remaining hypotheses.

## Materials and methods

Data for this analysis come primarily from two cross-national surveys conducted by the Pew Research Center's Religion and Public Life project. The first survey, *Tolerance and Tension: Islam and Christianity in Sub-Saharan Africa*, sampled more than 25,000 respondents in nineteen countries in sub-Saharan Africa in 2008–2009. The second survey, *The World's Muslims: Religion, Politics and Society*, sampled more than 30,000 Muslim respondents in twenty-six countries throughout the world in 2011–2012. The questionnaires for the surveys are very similar, although additional questions were added and some other questions were re-worded in the second survey. We focus our analysis only on questions that are consistent across both surveys.

We combine Muslim respondents from these surveys together to construct our full analytical dataset. Five sub-Saharan African countries (Democratic Republic of Congo, Botswana, Rwanda, South Africa, and Zambia) were excluded from analysis because of minuscule (2% or less) Muslim populations. Four other countries were excluded because key questions for the analysis were not asked in those countries (Iran, Morocco, Mozambique, and Uzbekistan). Finally, Thailand was excluded because the sample was not nationally representative. Fig 1 shows a map of the countries included in our analysis with shading by the percent Muslim in

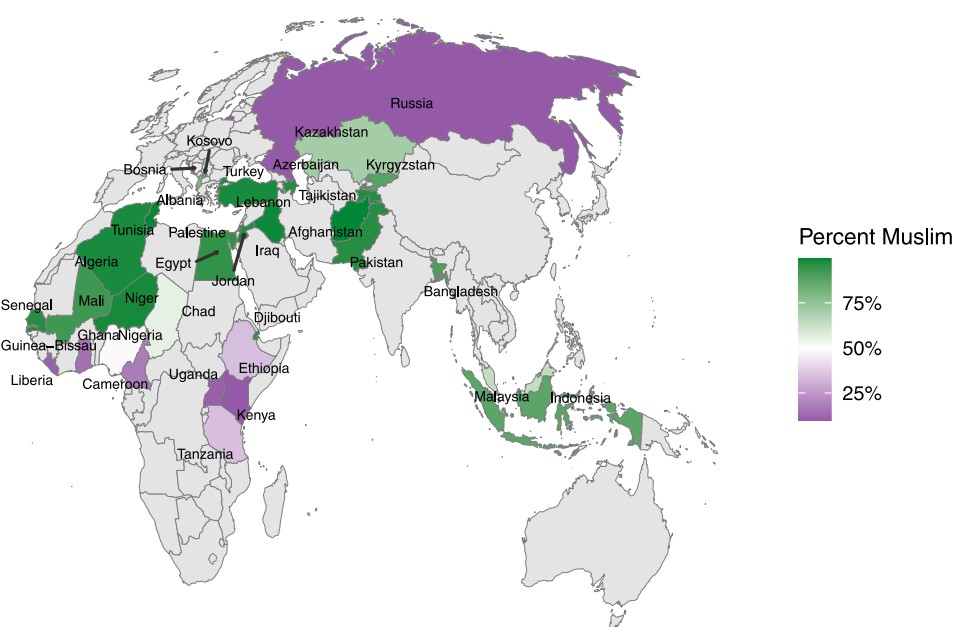

**Fig 1. Map showing countries included in the analysis by percent Muslim.** Mollweide projection used to preserve area proportions.

each country. The countries in the sample cover all regions of the world with substantial Muslim populations. In total, we use data from 35 countries and 35,400 respondents. The average sample size per country was 1,011 respondents with a minimum size of 245 respondents in Cameroon and a maximum size of 1,918 respondents in Bangladesh.

Because the country-level surveys were designed to be nationally representative, the data should be representative of within-country populations, but not necessarily the global Muslim population. Collectively, the countries in the sample roughly account for sixty-five percent of the global Muslim population [49], but some important countries both symbolically (e.g. Saudi Arabia, Iran) and by population weight (e.g. India) are absent.

For the dependent variable and several other variables in the analysis, we considered constructs made up of multiple individual items. We developed all constructs through an exploratory factor analysis. Furthermore, we tested both the reliability and measurement invariance of constructs across countries [50]. Because of the ordinal nature of all items involved in the construct, we use polychoric, rather than pearson, correlation to estimate the alpha measure of reliability and to conduct exploratory factor analysis [51]. Similarly, we use item response theory (IRT) models rather than multigroup confirmatory factor analysis to test for measurement invariance because IRT is better suited to analyze categorical ordinal outcomes [52]. Because frequentist test statistics are likely to identify even trivial deviations from invariance in a sample this large [50, p. 106], we use the Bayesian Information Criterion to evaluate IRT models that test sequentially for metric and scalar invariance.

## Measuring support for violent practices

We develop our measures of support for violent practices based on multiple questions that were fielded in both surveys. Respondents were asked yes/no questions for whether they favored (a) the death penalty for people who leave Islam, (b) harsh punishments like whipping and cutting off hands for crimes like theft and robbery, and (c) stoning people who committed

adultery. We also considered a likert-scale question on the justifiability of violence against civilians in defense of religion, but this variable had a much weaker correlation with the three other measures and so we dropped it from analysis. We constructed the dependent variable for our models by counting the number of favorable responses to each of the remaining questions and then re-scaling this tally to have a mean of zero and a standard deviation of one.

Fig 2 shows the average level of support by country for the three items that make up our measure of support for violent practices. Countries tend to have similar levels of support for all items within the construct, but the variation across countries is substantial. Support for these values ranges from virtually no support in countries such as Azerbaijan and Kazakhstan to nearly universal support in countries such as Afghanistan and Egypt.

Globally, these three measures were highly correlated with one another ($\alpha = 0.93$) and exploratory factor analysis indicated that a single scale worked well for all three variables. At the country level, $\alpha$ was above the standard cutoff for good reliability of 0.7 for all countries except for Jordan and Egypt, which still had reasonable $\alpha$ values of 0.68 and 0.62, respectively.

When testing for measurement invariance, we found some evidence of metric invariance. After analyzing model fit, we found that this metric invariance was caused by low correlations between the death for apostasy variable and the other two variables in Egypt and Jordan. Removing either of these countries led to a model fit that supported metric invariance. Fig 2 shows that the ordering of overall support for each item is different in Egypt and Jordan than in the remaining countries. In Egypt and Jordan, respondents were substantially more supportive of death for apostasy than the other two measures, whereas in the remaining countries, this item typically has similar or lower support. These results suggests that as support for this item approaches universality in these countries, its correlation with remaining items declines. Previous work has noted that this issue is a common problem in analyzing measurement

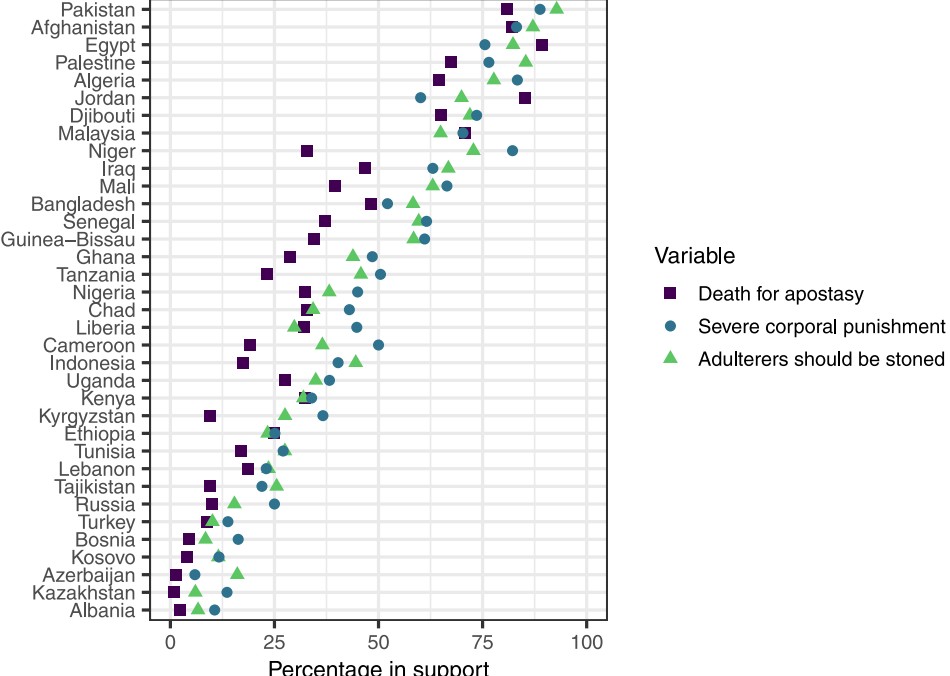

**Fig 2. Dotplot of percentage of Muslim respondents who support three different questions measuring support for violent practices for the violation of norms.** Countries are ordered from lowest to highest average level of support across all three questions.

invariance when group-level item means vary across the entire range of the distribution [53]. We have decided to retain Egypt and Jordan in the full analysis, but some caution should be taken in interpreting the specific results for these countries.

Further tests indicated scalar invariance in this measure. Scalar invariance is an extremely common issue in cross-national research [54], but the literature currently offers little guidance in how to resolve such problems. While scalar invariance technically indicates that means across groups are not directly comparable, it may in some cases have a minimal impact on such comparisons. As a sensitivity test, we estimated separate multilevel logit models on each of the items within the construct and compared the fixed and random components of each model to one another and to the model using the construct itself. Overall, both the fixed and random components are highly consistent across all models and none of our substantial conclusions would be modified by use of the separate models. Importantly, the random intercepts across the separate logit models (which measure the mean value in a country) are highly correlated ($r > = 0.92$) with the random intercepts of the construct model in all three cases. Therefore, on the principle of parsimony, we use the construct-based model for the remainder of the analysis here. The results of the sensitivity analysis are provided in the supplementary materials.

## Measuring development

At the country-level, we considered two different possibilities for a measure of overall development. Gross domestic product (GDP) per capita is often used as a purely economic measure of the level of development. On the other hand, the Human Development Index (HDI), produced by the United Nations Development Program, offers a more holistic measure of development. The HDI considers three separate components in its overall measure: life expectancy at birth, overall educational attainment, and gross national income per capita. These three components are combined into a single composite index.

These two measures are highly correlated among the countries used in the analysis ($r = 0.78$). All models reported here use the HDI. We chose the HDI as our primary measure of development because it allows us to parsimoniously capture multiple dimensions of development. As a sensitivity analysis, we constructed identical models with a logged GDP per capita measure. The results were similar, although the logged GDP variable tended to produce slightly smaller overall effects than HDI. The models from this sensitivity analysis are available in the supplementary materials.

## Independent variables

Education is a key measure in our analysis, but the categories used for educational attainment are country-specific, making it challenging to include in a cross-national analysis. Educational attainment was reported as a three-category system for the sub-Saharan African countries and relative to country-specific educational systems for the remaining countries. These country-specific systems ranged from a high of twelve distinct categories to a low of five distinct categories. We harmonized these measures by creating a quantile score based on the distribution of the educational attainment variable within each country. Each respondent was assigned a mid-point value on the quantile score between their given level of education and the next lowest level (or zero when the respondent was in the lowest category). This quantile score measures a respondent's relative position on the distribution of education within a country. We also considered a three-category measure of absolute educational attainment consisting of no secondary education, at least some secondary education, and post-secondary education that could be used for all countries. Sensitivity tests showed that these models produced similar results, but

the model fit statistics strongly preferred the educational quantile measure. Full results are available in the supplementary materials.

We measure each respondent's religiosity using a concept that combines the experiential and ritualistic dimensions of Glock and Stark's framework [55]. Our goal is to isolate the importance of religion in a person's daily life as measured by their personal experience and practice of it from specific ideological beliefs and knowledge about the religion. We considered a three-item construct including a question on the importance of religion in a person's life, the frequency of mosque attendance, and the frequency of prayer. Across the entire sample, this three item construct had a reasonable $\alpha$ of 0.78. However, country-level $\alpha$ values for this religiosity construct varied considerably and fourteen countries had an $\alpha$ below 0.6. Therefore, we decided to include each of the three variables measuring religiosity separately into the models. As the results below will show, each variable had a similar average effect on the outcome variable, but the pattern of random country-level slopes was substantially different for each variable.

Prior research has shown that religiosity is positively associated with theological literalism, conservativism, and fundamentalism among Muslims [11]. Individuals with more literal and theologically conservative beliefs are likely to have more anti-liberal values simply because some of the measures used for the dependent variable can be argued to have scriptural justification. To control for this potential confounding influence in our analysis, we tested a separate construct that measured theological conservatism and literalism in a person's actual beliefs. We considered questions on whether a person must believe in god to be moral, whether Islam is the one true faith, and whether there was only one way to interpret religious teachings. However, these questions did not hold up well as a single construct ($\alpha$ = 0.56). Therefore, to capture theologically conservative beliefs and practices, we included all three questions as separate independent variables in our models.

We include a variety of other individual-level variables in all models. We include variables for the age (five-year brackets), gender, income, and urbanicity of the respondent. Income was reported in loosely defined quartiles (i.e. four categories from "low" to "high") for the sub-Saharan African countries and brackets that were specific to each national currency for remaining countries and so we use a method identical to that for education to create income quantile scores. The country-specific income brackets ranged from a high of seventeen categories to a low of five categories.

We also include a measure of the respondent's self-reported denomination. The primary division here is between Sunni and Shia, but many respondents also identified in a non-denominational way (e.g. "just a Muslim") which we retain as a separate category. We combine all other denominations into a single "other" category. A separate item asked respondents whether they identified with any Sufi orders. We code this response as a binary variable that is distinct from the denominational categories.

We also control for non-religious ideological views. We constructed two indices of socially conservative views based on the respondent's answers regarding the acceptability of several stigmatized behaviors. Exploratory factor analysis suggested that the best fit was provided by a two factor solution that divided the items into those associated with death (euthanasia, suicide, abortion; $\alpha$ = 0.82) and those associated with sex (prostitution, homosexuality, and premarital sex; $\alpha$ = 0.88). This two factor solution also was metrically invariant across countries, although not scalar invariant. We also include two questions that gauge respondents' attitudes toward Western movies, music, and television. Finally, we include questions on whether religion is in conflict with modernity, and whether the country would be better off with a strong leader rather than a democratic government.

## Multiple imputation

We use multiple imputation to address missing values in the dataset. We imputed five complete datasets using multivariate imputation by chained equations [56]. We generally did not impute values when a question was missing because it was not asked in a particular country, but we made two exceptions to this rule. Questions regarding the moral acceptability of prostitution, premarital sex, and homosexuality were not asked in Afghanistan and the two questions about the morality of western movies, music, and television were not asked in Lebanon. In both of these cases, we imputed values in order to maximize country sample size. The percentage of values imputed ranged from a high of 11.4% for the question on whether religion was in conflict with modernity to 0.7% for the respondent's age. We included dependent variables in the imputation procedure, but drop cases that were missing on the dependent variable. After this exclusion, the sample size of our analytical dataset is 31,528 respondents.

## Scaling variables

To facilitate comparisons between variables in the models, all quantitative variables have been re-centered on the mean and re-scaled. The dependent variable has been divided by its standard deviation. Independent variables have been divided by twice their standard deviation. The estimated effects of quantitative variables shown throughout this article can be interpreted as the expected change in standard deviations of the dependent variable for an increase of two standard deviations in the independent variable. Dividing by twice the standard deviation makes the slopes of quantitative variables in linear models roughly comparable to the effects of categorical dummy variables [57].

## Results

Table 1 shows the results of three multilevel models predicting support for violent practices. Model 1 includes all independent variables, but no interaction term. Model 2 interacts the country-level HDI measure with the individual-level education and religiosity measures. Based on the results of Model 2, Model 3 removes the interaction between HDI and religiosity.

All of the independent variables are highly consistent across models. Although we focus primarily on the effects of education, religiosity, and development, we first briefly summarize the effects of other variables on support for violent practices. Income has little effect on support for violent practices. Similarly, we observe no difference between men and women in support for violent practices. Differences across age generally show that the youngest cohorts (under age 35) are more supportive of violent practices than older cohorts. Urban residents are less likely to support violent practices than those in rural areas. In terms of denomination, Sunnis are the most supportive of violent practices and Shias the least supportive, with other denominations and "just a Muslim" respondents falling in between. Self-identifying as a member of a Sufi order is associated with greater support for violent practices. Theological conservatism is generally associated with greater support for violent practices, but the strongest effect among these items is the belief that Islam is the one true faith. Respondents with more anti-Western views were also more supportive of violent practices. Preferences for democracy has little to no association with support for violent practices. Social conservatism is not positively associated with support for violent practices, and may in fact be negatively associated.

The three measures of religiosity are all strongly positively correlated with support for violent practices. In fact, frequency of prayer had the strongest association with support for violent practices among the individual-level variables and the other two measures of religiosity also exhibited some of the strongest associations. This strong association is observed even when holding constant a variety of demographic and ideational variables. On average, across

**Table 1. Results from multilevel models predicting support for violent practices for violating norms among Muslims.**

| | Model 1 | Model 2 | Model 3 |
|---|---|---|---|
| Intercept | -0.205 (0.084)* | -0.196 (0.085)* | -0.200 (0.084)* |
| Mosque attendance | 0.111 (0.022)*** | 0.107 (0.023)*** | 0.107 (0.022)*** |
| Importance of religion | 0.089 (0.032)** | 0.084 (0.033)* | 0.084 (0.032)** |
| Frequency of prayer | 0.155 (0.034)*** | 0.164 (0.034)*** | 0.164 (0.034)*** |
| Education quantile | -0.060 (0.022)** | -0.057 (0.021)** | -0.057 (0.021)** |
| Income quantile | -0.009 (0.010) | -0.009 (0.010) | -0.009 (0.010) |
| Age (ref. 18–24) | | | |
| Age 25–29 | 0.006 (0.014) | 0.006 (0.014) | 0.006 (0.014) |
| Age 30–34 | -0.008 (0.015) | -0.008 (0.015) | -0.008 (0.015) |
| Age 35–39 | -0.043 (0.016)** | -0.043 (0.016)** | -0.043 (0.016)** |
| Age 40–44 | -0.054 (0.016)*** | -0.054 (0.016)** | -0.054 (0.016)** |
| Age 45–49 | -0.033 (0.018) | -0.033 (0.018) | -0.033 (0.018) |
| Age 50–54 | -0.047 (0.019)* | -0.047 (0.019)* | -0.047 (0.019)* |
| Age 55–59 | -0.061 (0.021)** | -0.060 (0.021)** | -0.060 (0.021)** |
| Age 60 and over | -0.025 (0.019) | -0.024 (0.019) | -0.024 (0.019) |
| Female | -0.011 (0.010) | -0.011 (0.010) | -0.011 (0.010) |
| Urban | -0.048 (0.009)*** | -0.048 (0.009)*** | -0.048 (0.009)*** |
| Denomination (ref. Sunni) | | | |
| Shia | -0.130 (0.020)*** | -0.129 (0.020)*** | -0.129 (0.020)*** |
| Other denomination | -0.050 (0.030) | -0.049 (0.030) | -0.049 (0.030) |
| Just a Muslim | -0.087 (0.012)*** | -0.087 (0.012)*** | -0.087 (0.012)*** |
| Sufi (ref. non-Sufi) | 0.064 (0.014)*** | 0.065 (0.014)*** | 0.065 (0.014)*** |
| Believe in god to be moral | 0.035 (0.012)** | 0.036 (0.012)** | 0.036 (0.012)** |
| Islam is the one true faith | 0.150 (0.013)*** | 0.150 (0.013)*** | 0.150 (0.013)*** |
| One way to interpret relig. teachings | 0.015 (0.010) | 0.015 (0.010) | 0.015 (0.010) |
| Religion in conflict with modernity | 0.077 (0.010)*** | 0.076 (0.010)*** | 0.076 (0.010)*** |
| Dislikes western culture | 0.119 (0.010)*** | 0.119 (0.010)*** | 0.119 (0.010)*** |
| Western culture is immoral | 0.034 (0.010)*** | 0.034 (0.010)*** | 0.034 (0.010)*** |
| Prefers strong leader to democracy | 0.018 (0.010) | 0.018 (0.010) | 0.018 (0.010) |
| Socially conservative scale, death | -0.034 (0.011)** | -0.034 (0.011)** | -0.034 (0.011)** |
| Socially conservative scale, sex | -0.011 (0.011) | -0.010 (0.011) | -0.011 (0.011) |
| Human development index (HDI) | -0.330 (0.130)* | -0.238 (0.169) | -0.279 (0.134)* |
| HDI x Mosque attendance | | 0.002 (0.046) | |
| HDI x Importance of relig | | 0.031 (0.066) | |
| HDI x Frequency of prayer | | 0.014 (0.070) | |
| HDI x Education quantile | | 0.077 (0.042) | 0.079 (0.029)** |
| N (individual) | 31528 | 31528 | 31528 |
| N (country) | 35 | 35 | 35 |
| BIC | 72173 | 72224 | 72182 |

***$p < 0.001$;

**$p < 0.01$;

*$p < 0.05$

Table Notes: All models include random country-level intercepts and slopes for some variables. All quantitative variables are divided by twice their standard deviation for comparability. Results are based on five complete datasets with imputation for missing values.

countries, Muslims who are more religiously observant and devout are more likely to support violent practices and this result is not simply a function of demography, theological conservatism, social conservatism, or anti-Western sentiment.

The education quantile measure has a substantial negative association with support for violent practices, but the magnitude here is much smaller than for the religiosity variables. A two standard deviation increase in the education quantile is associated with roughly a 0.06 standard deviation reduction in the support for violent practices scale. The results are inconsistent with the expectations of Huntington expressed in *H2d*.

The results for the HDI measure in Model 1 also shows that the level of development in a country is strongly negatively associated with support for violent practices in that country. A two standard deviation increase in the HDI is associated with a 0.330 standard deviation decrease in a country's average on the support for violent practices scale. Individuals from more developed countries are much less supportive of violent practices, on average, as expected by *H1*.

In Model 2, we allow for an interaction between HDI and our measures of religiosity and education. Effectively, we are allowing the association between religiosity/education and support for violent practices to vary by the level of HDI, as predicted by *H2b* and *H3b*. There is little evidence of variation in the effect of any of the religiosity variables by HDI, casting doubt on *H3b*. The effect of education on support for violent practices, on the other hand, does vary substantially by HDI, but in a direction opposite to the expectations of *H2b*. The negative association between education and support for violent practices becomes weaker in more developed countries, rather than stronger. In the most developed countries, the education association may even reverse direction. While this result is not statistically significant in Model 2, Model 3 shows that the lack of significance is due to multicollinearity problems in Model 2 that inflated the standard error.

We now turn to the random country effects estimated by the models. The variance and correlation of the random intercepts and slopes is shown in Table 2. All of the slopes varied substantially across countries, but the largest variation was for the questions on whether religion is important and the frequency of prayer.

To explore this variation in more detail, we plot the random slopes for all countries in Fig 3. This figure clearly shows the substantial variation in the effects of these variables across countries. As expected, the religiosity slopes are positive and the education slopes are negative for most countries. However, we do observe a small number of countries with estimated slopes in the opposite direction for all four variables. The number of countries with negative slopes for mosque attendance and frequency of prayer is small and the magnitude of these negative slopes is small. For the question on whether religion is important, we see more negative effects

**Table 2. Standard deviation and correlation between random intercepts and slopes for religiosity and education.**

|  | SD | Correlation | | | |
|---|---|---|---|---|---|
|  |  | Intercept | Mosque attendance | Religion important | Frequency of prayer |
| Intercept | 0.488 |  |  |  |  |
| Mosque attendance | 0.099 | -0.253 |  |  |  |
| Religion important | 0.163 | 0.188 | 0.116 |  |  |
| Frequency of prayer | 0.177 | 0.629 | -0.380 | -0.218 |  |
| Educational quantile | 0.108 | 0.429 | -0.656 | -0.515 | 0.758 |

Note:

Correlation coefficients based on Model 3 of Table 1.

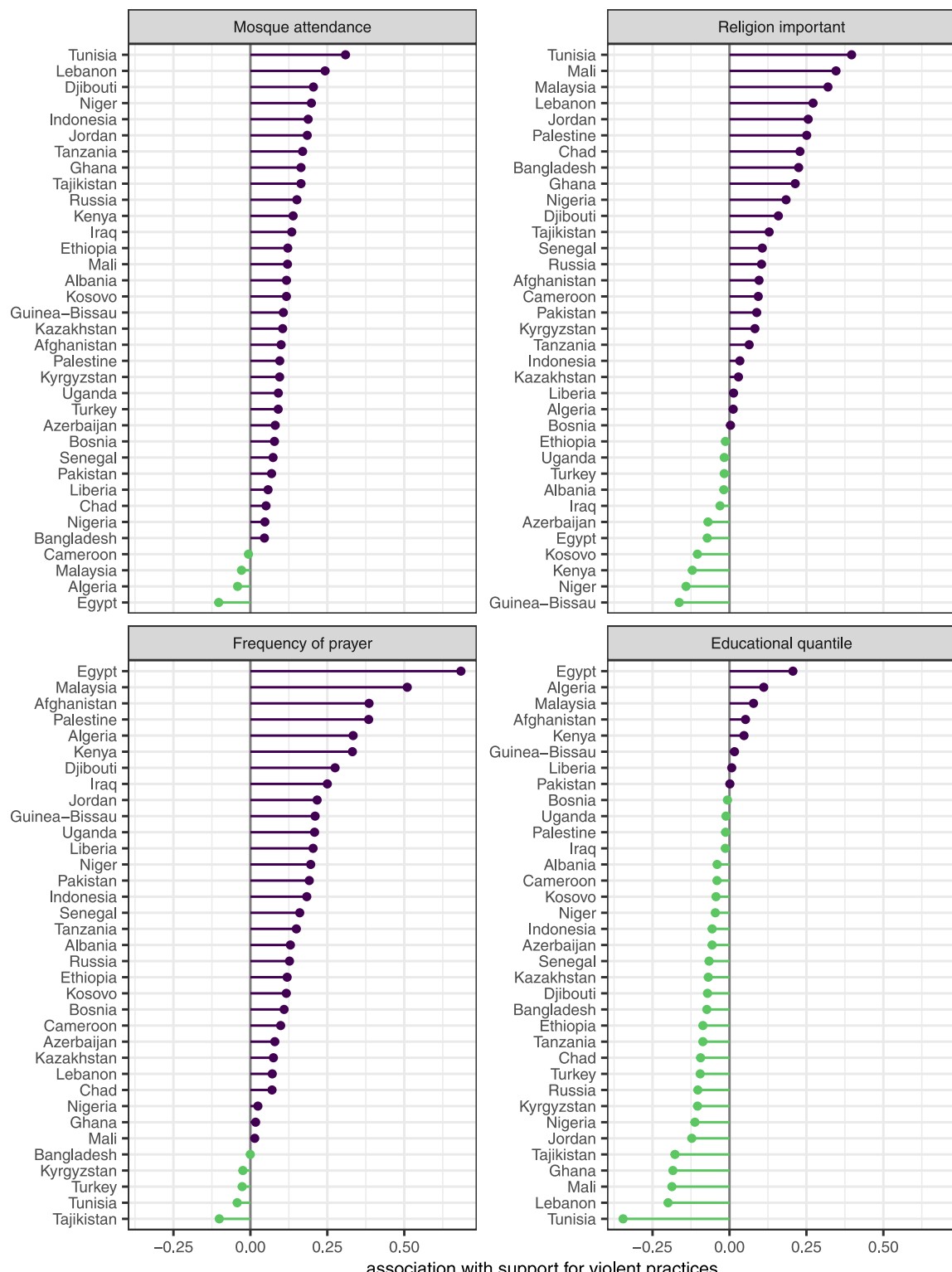

**Fig 3. Lollipop plots of the association between religiosity/education and support for violent practices in each country.** Values are based on random slopes from Model 3 of Table 1. Each panel is ordered from largest to smallest association. Values are color-coded by direction.

and a few (Kenya, Niger, Guinea-Bissau, and Kosovo) that are more substantial in magnitude. In these countries, individuals for whom religion is more important are less supportive of violent practices.

Eight countries (Egypt, Algeria, Malaysia, Afghanistan, Kenya, Guinea-Bissau, Pakistan, and Liberia) have estimated slopes that indicate a positive association between educational quantile and support for violent practices. The positive effects are negligible for Guinea-Bissau, Pakistan, and Liberia, but substantial for the remaining countries. In these countries, individuals with more education tend to be more supportive of violent practices.

Overall, both religiosity and education are highly variable in their effects across countries, casting doubt on *H2a* and *H3a* which expected largely consistent and similar effects across countries. While the average effects across countries are broadly consistent with the expectations of classic modernization theory, the variability across countries is not. We now turn to an exploration of that variability in more detail.

We begin by looking at the correlation of each of the random slopes with the random intercepts. These correlations tells us how the average level of support for violent practices in each country is related to the association of education/religiosity with that support. Table 2 shows the overall correlation coefficients between these variables, while Fig 4 plots these relationships in more detail using a scatterplot.

As Table 2 indicated, the country-specific education slopes are positively correlated with the average support for violent practices in a country ($r = 0.429$). In countries with less support for violent practices overall, the association between education and support for violent practices tends to be more negative. The correlation here is consistent with the expectation of *H2c* and indicates a substantial socialization effect on educational institutions. In more liberal countries (those with less support for violent practices overall), education has more of a liberalizing influence and in more conservative countries, education has less of a liberalizing effect, and may even have a conservatizing effect.

The correlation between each of the religiosity slopes and the average support in a country do not tell as consistent a story. For frequency of prayer, we observe a similar, and even stronger, positive relationship ($r = 0.629$). We find that in more liberal countries, frequency of prayer has a less conservatizing effect on support for violent practices. These results suggest a similar socialization effect as for education.

For the remaining religiosity variables, the correlations are much weaker and in opposite directions. We observe a positive association for the importance of religion variable ($r = 0.188$), indicating that there may be some socialization effect on this measure. The effect of mosque attendance, however, is negatively associated with the level of support in a country ($r = -0.253$). In more liberal countries, mosque attendance tends to be more positively associated with support for violent practices.

We finally turn to the correlation between the religiosity and education slopes themselves. Do the effects of religiosity and education within a country tend to work in synchronicity or are they polarized? As Table 2 shows, all three of these correlations are quite strong but do not operate in the same direction. Fig 5 further explores these relationships by drawing pairwise scatterplots between each of the religiosity random slopes and the educational quantile slopes. For both mosque attendance and the question on whether religion is important, we observe strong negative correlations ($r = -0.656$ and $r = -0.515$, respectively), consistent with *H4a*. These results indicate polarization between educational and religious institutions. In countries with the most negative association between education and support for violent practices, we also tend to find the most positive association between these religiosity measures and support for violent practices.

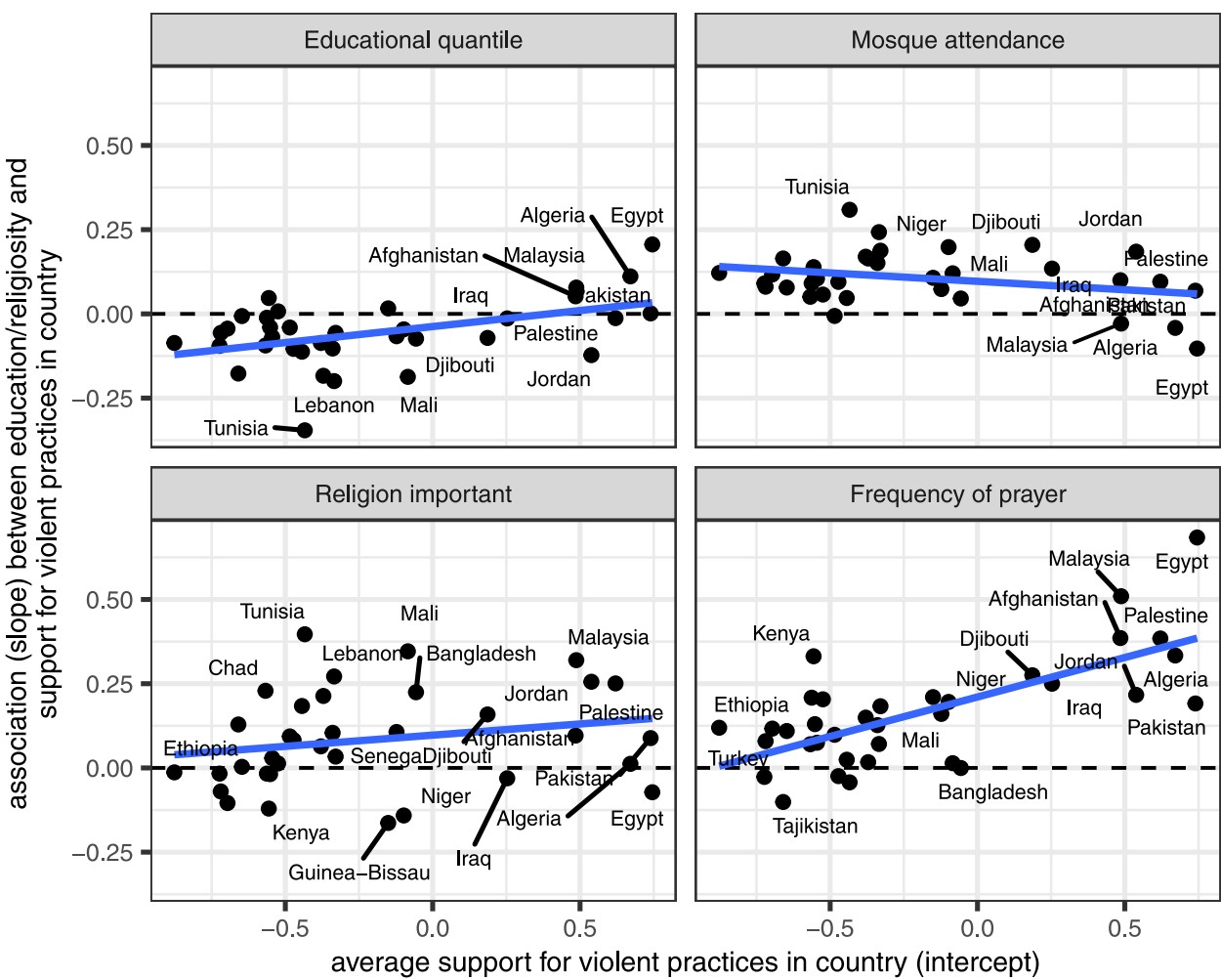

**Fig 4. Scatterplot showing the relationship betwen a country's average level of support for violent practices and the association between education/religiosity and support for violent practices in a country.** Values are based on random intercepts and slopes from Model 3 of Table 1. Best fitting OLS line is shown in each panel.

The frequency of prayer results tell a different story, however. The country-level slopes here are strongly positive correlated ($r = 0.758$), indicating synchronization consistent with *H4b*. The countries where education has the strongest negative association with support for violent practices also tend to be the countries where frequency of prayer has the smallest positive (and occasionally, negative) association with support for violent practices.

Overall, the results here reveal the relationship between religiosity and support for violent practices is particularly complex and depends on which aspect of religiosity we focus on. Overall, frequency of prayer seems the most responsive to other social pressures. The positive effect of frequency of prayer is substantially reduced and may even reverse direction in the most liberal countries and in countries where education has the strongest negative effects. In contrast, mosque attendance and the importance of religion seem quite stubbornly resistant to such social pressure. Neither case showed a strong correlation with overall support and both cases evidenced a strong polarization with regard to the effect of education.

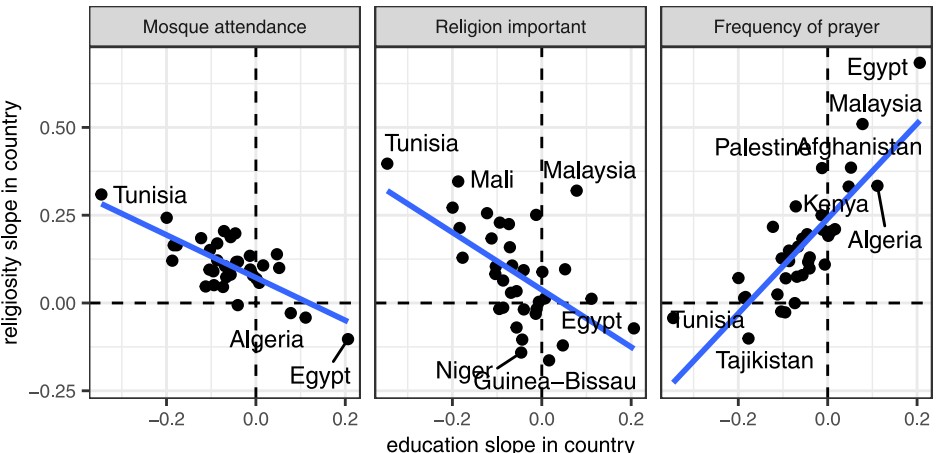

**Fig 5. Scatterplot showing the relationship between country-level associations of education and religiosity with support for violent practices.** Values based on random slopes from Model 3 of Table 1. Best-fitting OLS line is shown in each panel.

## Conclusion

In this article, we have analyzed variation in support for violent practices among Muslims in thirty-five countries, paying particular attention to the association of education, religiosity, and development with these attitudes, as well as the heterogeneity in this association across countries.

At first glance, the results seem consistent with the expectations of modernization theory. Greater development in a country, as measured by the Human Development Index, strongly predicted less support for violent practices. Similarly, when averaged across countries, the effects of education and religiosity are as expected from prior research in predominantly non-Muslim contexts. More educated individuals are moderately less supportive of violent practices while individuals with greater religiosity are substantially more supportive of violent practices. Furthermore, this religiosity association remains strong even after controlling for theological conservatism and a variety of other ideational variables.

However, modernization theory is not supported when we look more closely at the heterogeneity in the effects of education and religiosity on support for violent practices across countries. Both of these effects were highly variable across countries. In a minority of countries, the expected relationships were even reversed.

Furthermore, the association between education and violent practices deviated from the expectations of modernization theory in two ways. First, we found that the negative relationship between education and support for violent practices is weakest (and may even change direction) in the most developed countries rather than the least developed countries. Second, we found that the association between education and support for violent practices is the most negative when countries have less overall support for violent practices. This latter finding is more consistent with the socialization hypothesis than the development hypothesis in accounting for the relationship between education and liberal values. In short, we find no evidence of a universally consistent effect of education on values, but rather one mediated by social context.

The results for religiosity are somewhat more complex, owing to the fact that the patterns of country-level heterogeneity were different across the three variables we used to measure religiosity. The positive association of frequency of prayer with support for violent practices was

reduced in countries with less overall support for violent practices, suggesting that this form of religiosity was affected by socialization in a manner similar to education. However, the other two measures of religiosity were not strongly associated with the overall level of support for violent practices in a country. We also found that, for all three variables, the effect of religiosity on support for violent practices was not associated with the level of development in the country. Overall, religiosity seems considerably less responsive to social context than education, but this depends somewhat on which aspect of religiosity we choose to focus on.

We also explored how the variation in country-level education and religiosity effects might be related to one another. We identified strong correlations between these effects, but we again found that the results varied by which measure of religiosity we examined. The effects of mosque attendance and the importance or religion were strongly negatively associated with the effects of education. These results suggest polarization between educational and religious institutions within countries. Countries with the most negative association between education and support for violent practices tend to have the most positive association between mosque attendance/importance of religion and support for violent practices.

Interestingly, we found an opposite pattern of synchronization in the strong positive association between education effects and the effects of frequency of prayer. In this case, countries with the most negative association between education and support for violent practices tend to also have the least positive (and occasionally negative) association between frequency of prayer and support for violent practices. These results suggest that this particular form of religiosity is more amenable to a softening influence in the face of larger social change.

Overall, the heterogeneity we observe across countries in the effects of education and religiosity casts doubt on the universalistic assumptions of classic modernization theory. This heterogeneity also casts doubt on the utility of the civilizational argument to understand value change. We find no evidence of a singular civilizational pattern of Islamic societies. Rather, we find that the general patterns of Islamic "civilization" are consistent with those observed in non-Islamic societies but with substantial variation at the country level.

Instead, our results support a multiple modernities perspective and a socialization perspective. Social institutions respond differently to social change in different ways across countries and this produces heterogeneity in effects across countries. Some of this heterogeneity we can account for by observable social context, such as the overall level of development and overall support for violent practices. However, much of this variation remains unaccounted for and would need to be explored by a more detailed comparative analysis of the historical development of each country.

To highlight this last point, We draw attention to the results for Algeria and Tunisia. Algeria and Tunisia are arguably two of the most similar countries in our dataset by geography, language, and culture. Yet, the observed relationships between education and support for violent practices are not only opposite in these two countries, but represent two of the most extreme values in the data. In Tunisia, educational attainment strongly predicts less support for violent practices, while in Algeria, educational attainment strongly predicts greater support for violent practices. The implication is that we must delve deeper into the particular historical development of each country in order to understand this variation.

We draw two additional implications from our results that are relevant for related research. First, the heterogeneity across countries suggests that quantitative analysis of values and attitudes among Muslims using a small set of countries can easily draw misleading results depending on which countries are selected. Much of the prior research on this topic uses samples of five or fewer countries. Attempts to generalize from such samples to the global Muslim population may be suspect.

Second, the effects of religiosity on attitudes may depend on which measures one employs. We found very different patterns across the three measures of religiosity we used here and also detected issues of unreliability and invariance when trying to develop a construct from these measures. Prior work often relies on a single measure and this may mislead researchers about the potentially variable effects of different elements of religiosity.

Our results are not without their own weaknesses. Strong causal claims are limited by the cross-sectional nature of the data. For example, while we find that Muslims in countries with greater development tend to have less support for violent practices, we cannot show that a change in development is associated with a weakening of those attitudes. Future longitudinal data would be valuable at strengthening our understanding of causal mechanisms, albeit diffi-cult and costly to obtain on a sample of this size and scale.

Furthermore, although our sample of countries is quite large, it is still a non-random sam-ple and could potentially misrepresent global patterns. In particular, our sample is missing two countries, Saudi Arabia and Iran, which are symbolically very important in discussions of Islamic civilization.

The goal of this article has been to use statistical analysis to leverage a large dataset and pro-vide a birds-eye view of the relationship between support for violent practices and individual and country-level predictors. By necessity, this approach ignores the specificity and nuance of country-specific historical development. Our intent is not to discount or replace work that focuses on particular historical cases or small N comparative/historical analyses, but rather to complement this work with a broader comparative approach. Indeed one of the central find-ings of our analysis is the substantial variation in key relationships between countries. This suggests the possibility of fruitful explorations at the country-level using multiple methodolog-ical approaches to understand the substantial heterogeneity documented here.

## Supporting information

**S1 Appendix. Supplementary materials.** This appendix contains tables of models using alter-nate specifications of religiosity, education, and development.
(PDF)

## Acknowledgments

The data analyzed for this article were collected and distributed by the Pew Research Center. Pew Research Center bears no responsibility for the analyses or interpretations of the data pre-sented here. The opinions expressed herein, including any implications for policy, are those of the authors and not of Pew Research Center.

## Author Contributions

**Conceptualization:** Aaron Gullickson, Sarah Ahmed.

**Data curation:** Aaron Gullickson, Sarah Ahmed.

**Formal analysis:** Aaron Gullickson, Sarah Ahmed.

**Methodology:** Aaron Gullickson, Sarah Ahmed.

**Project administration:** Aaron Gullickson.

**Writing – original draft:** Aaron Gullickson, Sarah Ahmed.

**Writing – review & editing:** Aaron Gullickson, Sarah Ahmed.

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
