## [Decision Letter · Decision Letter 0]

25 Jun 2021

PONE-D-21-14799

The role of education, religiosity and development on support for violent practices among Muslims in thirty-five countries

PLOS ONE

Dear Dr. Gullickson,

Thank you for submitting your manuscript to PLOS ONE. After careful consideration, we feel that it has merit but does not fully meet PLOS ONE’s publication criteria as it currently stands. Therefore, we invite you to submit a revised version of the manuscript that addresses the points raised during the review process.

Two scholars of research on your topic see substantial merit in your work. I concur.  The reviewers offer you some very helpful advice in revising the paper.  In my read of your article, and in response to Reviewer 2's concern that your findings are consistent with a socialization hypothesis but inconsistent with theories of modernization, perhaps you might consider engaging this point by drawing on Durkheimian and neo-Durkheimian theories of socialization and collective consciousness. 

We look forward to receiving your revised manuscript.

Kind regards,

Bryan L. Sykes, Ph.D.

Academic Editor

PLOS ONE

Journal Requirements:

Additional Editor Comments (if provided):

Reviewers' comments:

Reviewer's Responses to Questions

**Comments to the Author**

1. Is the manuscript technically sound, and do the data support the conclusions?

Reviewer #1: Yes

Reviewer #2: Yes

2. Has the statistical analysis been performed appropriately and rigorously? 

Reviewer #1: Yes

Reviewer #2: Yes

3. Have the authors made all data underlying the findings in their manuscript fully available?

Reviewer #1: Yes

Reviewer #2: Yes

4. Is the manuscript presented in an intelligible fashion and written in standard English?

Reviewer #1: Yes

Reviewer #2: Yes

5. Review Comments to the Author

Reviewer #1: Thanks for the opportunity to review this paper. I have found the paper well-written and well-organised. I have a few comments that might strengthen the paper:

• The introduction is strong and has started interestingly. There is a paragraph at the end of the introduction (lines 55-66) that presents the findings of this research. I think this should move to the later section, such as findings or conclusion.

• The authors made a great argument on “Heterogeneity and multiple modernities” and how we should be cautious in analysing the impact of education and urbanisation and Islamic fundamentalism. My question is that why did not authors consider the Postmodern approach?

• The Method has been explained very well, and detail has been provided appropriately.

• Conclusion and discussion: I wonder if you have found any previous literature to compare your findings. This is particularly important to provide evidence in the areas that you have found conflicting findings (education, for instance). There have been some discrepancies in the levels of education and support for violent practices. What do the other report tell us? Is there any possible explanation for this based on the current literature?

• The authors acknowledged that Iran and Saudi Arabia have been missing from this database in the Method section. This should be again discussed at the end of the paper because these two countries are very critical in analysing Muslim countries. This is due to several factors, including having fundamentalist governments, strong religious institutions and representing Sunni and Shia ideologies. What should be done in the future to get data from these two countries?

• What are the implications based on this paper’s findings and gaps?

Reviewer #2: There are many valid reasons to finally decide to publish this contribution, however, only after a number of improvements are to be included / added to the paper, as it stands now. These reasons for their innovative approach are spelled out correctly on p. 3 and actually contains the goal and new contributions of the paper which I strongly endorse. The first improvement I would like to suggest could be to explicitly phrase the research questions that guide this paper, including not only the main effects / relationships but also the cross-level interaction effects that will be addressed.

The paragraph on education and religiosity sums up relevant research which, however, could be improved by explicitly phrasing the hypotheses that follow from this previous research. The same goes for the paragraph on heterogeneity. Once these hypotheses are explicated, they can be directly related to the respective paragraphs in the analytical approach, starting on p. 5-6, providing a much more insightful and less technical summary of expectations as related to model parameters. If so, they could also add the equations expressing the cross-level interactions as described on p. 6. This contribution, however, could certainly be improved if the paragraph on ‘Background’ and Analytical approach and theoretical expectations’ would be more integrated and hence re-ordered, from simple direct relationships on the individual level, to direct relationships from the macro to the individual level, closing up with cross-level interactions.

In the paragraph on materials and methods, the authors claim that they developed all constructs using exploratory factor analysis. However, it is (at least) controversial to run factor analyses on nominal variables, like the core dependent variable. Moreover, the empirical evidence should be improved that these items related to the DV are not only reliable (expressed by Cronbach’s alpha) in each and every country, but also cross-nationally equivalent, and moreover, valid (see Leung and van de Vijver, 1997). For this purpose, the authors could apply probabilistic scalogram analysis, also known as Mokken scale analysis (Mokken, 1971; Sijtsma & Meijer, 2016; Sijtsma & Molenaar, 2002) that allows for nominal variables to be tested on dimensionality, and moreover, on cross-national equivalent validity. A similar improvement could be worthwhile considering is the cross-national equivalence and validity of the measurement of religiosity, which given the interval nature of these variables, could be tested with factor analyses for separate countries and for the pooled data.

Also in the Results section, several improvement are possible. To start with Table 1: it is useful to distinguish between interval versus categorical variables. The interval variables are just fine as they stand, however, the categorical variables should be labeled as: age, its categories and make explicit which one of these is the reference category. Particularly in the case of the variable denomination this is of importance: I guess that the Suni are the reference category and the parameters of the other denominations are to be described as differences from this reference category (e.g. Lewis-Beck, Applied Regression) which is obscured and incorrectly described in the text on p. 10: it is not true that the ….Sunis are the most supportive of violent practices…however, the Suffis are more supportive than the ref cat (+.073***).

I think that the authors come up with a plausible explanation as to why there is a positive relationship between education and the DV’s in some countries, stating that these findings are consistent with the socialization hypothesis. However, they are not consistent with theories on modernization, which should be also explained. This could be improved by actually addressing this theoretical debate. Overall, however, their account of the findings in the Conclusion paragraph is well balanced.

6. PLOS authors have the option to publish the peer review history of their article (what does this mean?). If published, this will include your full peer review and any attached files.

Reviewer #1: **Yes: **Dr Rojan Afrouz

Reviewer #2: No

---

## [Author Response · Author response to Decision Letter 0]

16 Sep 2021

A full response memo has been included in the PDF. We include that information here as well, but better formatting as well as a figure are available in the PDF.

---

We thank the editor and the reviewers for their thoughtful comments on the previous version of this manuscript. We have attempted to address all of the comments provided thoroughly and sincerely in this revised version of the manuscript. We feel that the manuscript is much improved as a result of these changes.

The most substantial changes to the manuscript are a result of Reviewer 2’s comments regarding testing the reliability and invariance of our constructs across countries. As a result of these tests, we decided to separate the religiosity construct into three separate variables. This change had some consequences for our results because we observe some differences in how the effects of these religiosity variables are correlated across countries. While the overall results are similar, we think that this finding is an important part of our results and we thank the reviewer for pushing us on this issue.

Below we discuss in detail how we addressed each reviewer’s concerns, point by point. Reviewer comments are indicated by ">".

Reviewer 1

> Thanks for the opportunity to review this paper. I have found the paper well-written and well-organised. I have a few comments that might strengthen the paper:

> The introduction is strong and has started interestingly. There is a paragraph at the end of the introduction (lines 55-66) that presents the findings of this research. I think this should move to the later section, such as findings or conclusion.

This may reflect differences in disciplinary style, as it is common in our discipline to highlight major findings at the bottom of the introduction section. To avoid too much duplication, we have shortened this paragraph and moved details of the findings to the conclusions. 

> The authors made a great argument on “Heterogeneity and multiple modernities” and how we should be cautious in analysing the impact of education and urbanisation and Islamic fundamentalism. My question is that why did not authors consider the Postmodern approach?

Unfortunately, we are a little uncertain about the reviewer’s intent in this case. They may be referring to postmodern (or relatedly “post-structuralist”) theories of social change (e.g. Foucault) or to postmodern societies as opposed to modern ones. Since much of the debate in the developing world still hinges on some notion of modernity, rather than postmodernity, we maintain our focus on modernization. However, we note here that some might consider the “multiple modernities” approach itself to be more consistent with a postmodern theoretical approach, given its focus on (sometimes unbounded) heterogeneity.

> The Method has been explained very well, and detail has been provided appropriately.

Thank you for the kind words. This section has been revised somewhat to address reviewer 2’s concerns but is largely the same. 

> Conclusion and discussion: I wonder if you have found any previous literature to compare your findings. This is particularly important to provide evidence in the areas that you have found conflicting findings (education, for instance). There have been some discrepancies in the levels of education and support for violent practices. What do the other report tell us? Is there any possible explanation for this based on the current literature?

This is an excellent point. We cannot compare our results directly to prior work because, to our knowledge, no prior work has looked specifically at the outcome variable that we analyze here. However, we do develop our expectations and compare our resultsmore generally to prior work on the relationship between education/religiosity and liberal values. In the revised manuscript, we try to more clearly identify how our results relate to that prior work - most notably that our findings are consistent with the socialization hypothesis. The changes that we have made to more formally state hypotheses in response to Reviewer 2 should also make these connections to prior work more clear. 

> The authors acknowledged that Iran and Saudi Arabia have been missing from this database in the Method section. This should be again discussed at the end of the paper because these two countries are very critical in analysing Muslim countries. This is due to several factors, including having fundamentalist governments, strong religious institutions and representing Sunni and Shia ideologies. What should be done in the future to get data from these two countries?

Since this data is secondary data from the Pew Research Center, we do not have direct knowledge of the difficulties in collecting data in these countries. We note on pg. 7 of the revised draft that Iran (along with three other countries) was included in the original Pew data but was missing data on several key variables and so we were forced to drop it. 

In addition to mentioning this lack in the material and methods section, we now include this as a limitation of the study in the penultimate paragraph on pg. 20. This paragraph reads: “Furthermore, although our sample of countries is quite large, it is still a non-random sample and could potentially misrepresent global patterns. In particular, our sample is missing two countries, Saudi Arabia and Iran, which are symbolically very important in discussions of Islamic civilization.” 

> What are the implications based on this paper’s findings and gaps?

Thank you for encouraging us to more explicitly spell out these implications. We have now explicitly added three implications to the conclusion section (pg. 19) that may help in guiding future research. Briefly, these implications are that:

1. Country-level heterogeneity matters for substantive conclusions and cannot be easily explained by cultural and regional similarities. Further work that focuses on detailed and deep comparative analysis of historical development is important. We highlight the cases of Tunisia and Algeria to make this point.

2. Country-level heterogeneity matters methodologically. Quantitative studies with a small sample of countries (which are common) could easily be misleading if they are used to generalize to Muslim populations, writ large.

3. The effects of religiosity seem to differ depending on which measure we choose to focus on. Studies that use only a single measure of religiosity (often due to data limitations) may mislead about the complexity of the effect of religiosity on values.

Reviewer 2

> There are many valid reasons to finally decide to publish this contribution, however, only after a number of improvements are to be included / added to the paper, as it stands now. These reasons for their innovative approach are spelled out correctly on p. 3 and actually contains the goal and new contributions of the paper which I strongly endorse. The first improvement I would like to suggest could be to explicitly phrase the research questions that guide this paper, including not only the main effects / relationships but also the cross-level interaction effects that will be addressed.

Thank you for this suggestion which really helped us in re-organizing the revised paper. We added four research questions at the bottom of the introduction (pg. 3). These questions are:

1. What is the relationship between support for violent practices and education, religiosity, and level of development across all countries?

2. How does the level of development in a country affect the relationship between support for violent practices and education/religiosity?

3. How is the average level of support for violence in a country associated with the relationship to education/religiosity in that country?

4. How are the country-specific effects of education and religiosity on support for violent practices associated with one another?

We also provided more information in the introduction regarding multiple modernities in order to help motivate these questions.

> The paragraph on education and religiosity sums up relevant research which, however, could be improved by explicitly phrasing the hypotheses that follow from this previous research. The same goes for the paragraph on heterogeneity. Once these hypotheses are explicated, they can be directly related to the respective paragraphs in the analytical approach, starting on p. 5-6, providing a much more insightful and less technical summary of expectations as related to model parameters. If so, they could also add the equations expressing the cross-level interactions as described on p. 6. This contribution, however, could certainly be improved if the paragraph on ‘Background’ and Analytical approach and theoretical expectations’ would be more integrated and hence re-ordered, from simple direct relationships on the individual level, to direct relationships from the macro to the individual level, closing up with cross-level interactions.

Thank you for these keen suggestions which have helped us re-organize these sections comprehensively. We have moved all of the theoretical expectations that were in the “Analytical Approach” section into the “Background” section and phrase them as explicit hypotheses. The “Background” section itself has been restructured somewhat to accommodate this by bringing the discussion of multiple modernities forward to justify the later hypotheses. The “Analytical Approach” section then focuses more explicitly on how the multilevel model parameters relate directly to the formal hypotheses. 

> In the paragraph on materials and methods, the authors claim that they developed all constructs using exploratory factor analysis. However, it is (at least) controversial to run factor analyses on nominal variables, like the core dependent variable. Moreover, the empirical evidence should be improved that these items related to the DV are not only reliable (expressed by Cronbach’s alpha) in each and every country, but also cross-nationally equivalent, and moreover, valid (see Leung and van de Vijver, 1997). For this purpose, the authors could apply probabilistic scalogram analysis, also known as Mokken scale analysis (Mokken, 1971; Sijtsma & Meijer, 2016; Sijtsma & Molenaar, 2002) that allows for nominal variables to be tested on dimensionality, and moreover, on cross-national equivalent validity. A similar improvement could be worthwhile considering is the cross-national equivalence and validity of the measurement of religiosity, which given the interval nature of these variables, could be tested with factor analyses for separate countries and for the pooled data.

The reviewer addresses two issues with our use of exploratory factor analysis to create constructs used for the analysis. The first issue is the use of techniques designed for quantitative variables using binary and polytomous categorical variables. The second issue is the robustness of these constructs across all of the countries in our dataset. The most important construct we use is the one for the dependent variable which is made up of three yes/no questions, but the religiosity construct is also important. The religiosity variable is made up of three ordinal variables with varying numbers of categories. 

The reviewer is correct that treating categorical variables as quantitative scores is potentially problematic both for calculating Cronbach’s alpha and for exploratory factor analysis. To address this in the revision, we have followed the currently prevailing advice in the literature to use polychoric correlation rather than pearson correlation to calculate the alpha measure of reliability and to conduct factor analysis (Gadermann et. al. 2012). All of the measures in the revised manuscript rely upon polychoric correlation.

The second issue is more complex. The issue of measurement invariance has recently become a major area of concern across a wide variety of disciplines, most notably psychology and political science. To address this issue, we (1) calculated our alpha measure of reliability separately for every country, and (2) used item response theory models to test for metric and scalar invariance. We chose item response theory rather than the more popular multigroup confirmatory factor analysis approach because item response theory is more appropriate for categorical responses (Tay et. al. 2015). The two techniques are related to one another and both can be used to test for metric and scalar invariance. We discuss our updated methodological procedures on pg. 8 of the revised manuscript.

The results for the dependent variable were somewhat complex. The reliability of the measure held up for all countries in the analysis although two countries (Egypt and Jordan) had reliability in the 0.6-0.7 range. These two countries also caused problems for our test of metric invariance. When either of these countries were excluded, we found that the results for the remaining countries passed the test of metric invariance, but not when they were both included. The reason seems to be the unusually high level of support for death for apostasy in Egypt and Jordan, which is only weakly correlated with the other two measures.. Welzel et. al. (2021) argue that invariance can result when group means vary across the entire range of the distribution (as they do here) and that this does not accurately reflect the value of the construct. Thus, we have decided to leave Egypt and Jordan in the analysis, with proper cautions about taking the results for these two countries with some grain of salt.

The test of scalar invariance indicated some degree of scalar invariance. This has become commonplace in the literature and some argue that tests of scalar invariance are almost always bound to fail (Meitinger et. al. 2020). The consequences of this failure in terms of our multilevel model are far from clear. To test the sensitivity of our results, we conducted three separate multilevel logistic regression models on each of the outcomes in our construct, based on the fullest model in the analysis (Model 3 of Table 1). We then compared these results to each other and to the results for the construct model. The results (in terms of both the fixed effects and the correlation of random components) were highly consistent across all models and our substantive results would not be meaningfully changed if we used three separate models instead of one. The random intercepts across all of the models were correlated with r>0.92 in each case. Given the similarity of results, it seems reasonable to use the construct-based model as a data reduction tool on the grounds of parsimony. We discuss these issues on pp. 9-10 of the revised manuscript and provide the results of the three logistic regression models in the supplementary materials.

The religiosity construct held up less well, as discussed on pp. 10-11 of the revised manuscript. Although the global reliability measure was adequate (0.78), country-specific reliability failed in a number of countries as shown in the figure below (in PDF). We therefore decided to add each measure of religiosity into the model separately. The revised results showed that while these variables all had a similar average effect on support for violent practices across countries, the correlation in their effects across countries was very different, thus justifying the decision to treat them separately. Although it has made our results a bit more complicated to discuss, we feel that this is an important finding and have highlighted it in the revised results and conclusions.

We also found low country-level reliability of our westernization attitude variable and so included both questions separately into our models. We also found that the construct measure of social conservatism held up better to tests of invariance using a two factor solution, so we now include two separate measures of social conservatism into the model. This is discussed on pg. 11 of the revised manuscript.

> Also in the Results section, several improvement are possible. To start with Table 1: it is useful to distinguish between interval versus categorical variables. The interval variables are just fine as they stand, however, the categorical variables should be labeled as: age, its categories and make explicit which one of these is the reference category. Particularly in the case of the variable denomination this is of importance: I guess that the Suni are the reference category and the parameters of the other denominations are to be described as differences from this reference category (e.g. Lewis-Beck, Applied Regression) which is obscured and incorrectly described in the text on p. 10: it is not true that the ….Sunis are the most supportive of violent practices…however, the Suffis are more supportive than the ref cat (+.073***).

We thank the reviewer for these suggestions which we have now implemented in Table 1 of the revised draft. To save space, we do not provide reference categories for most binary contrasts, which can be logically inferred, but we do provide reference categories for all polytomous variables (age and denomination). 

We note that, based on the survey format, “Sufi” is not an actual denomination but comes from a separate question regarding whether the respondent belonged to any Sufi order. This was noted in the text of the original and on pg. 11 of the revised draft, but we see how this was confusing given the structure of the previous table. To avoid this problem, we now note the reference category for Sufi in Table 1, which we think will clarify its measurement.

> I think that the authors come up with a plausible explanation as to why there is a positive relationship between education and the DV’s in some countries, stating that these findings are consistent with the socialization hypothesis. However, they are not consistent with theories on modernization, which should be also explained. This could be improved by actually addressing this theoretical debate. Overall, however, their account of the findings in the Conclusion paragraph is well balanced.

The editor also drew attention to this issue. After consideration, we agree that we probably give modernization theory too much credit in the conclusion, given the (1) heterogeneity of results, (2) support for the socialization hypothesis for education, and (3) the fact that the liberalizing effect of education is weaker in more developed countries. We have now highlighted these issues in the conclusion (pp. 18-19) and have been much more critical of classic modernization theory. However, we do still note that the large negative effect of the HDI measure on support for violent practices is broadly consistent with modernization theory. 

Editor’s Comments

> In my read of your article, and in response to Reviewer 2's concern that your findings are consistent with a socialization hypothesis but inconsistent with theories of modernization, perhaps you might consider engaging this point by drawing on Durkheimian and neo-Durkheimian theories of socialization and collective consciousness.

We highlight the inconsistency with modernization theory as indicated in our response to reviewer 2’s final comment.

Bibliography

Gadermann, Anne M., Martin Guhn, and Bruno D. Zumbo. “Estimating Ordinal Reliability for Likert-Type and Ordinal Item Response Data: A Conceptual, Empirical, and Practical Guide.” Practical Assessment, Research and Evaluation 17, no. 3 (2012): 1–13. https://doi.org/10.7275/N560-J767.

Meitinger, Katharina, Eldad Davidov, Peter Schmidt, and Michael Braun. “Measurement Invariance: Testing for It and Explaining Why It Is Absent.” Survey Research Methods 14, no. 4 (October 10, 2020): 345-349 Pages. https://doi.org/10.18148/SRM/2020.V14I4.7655.

Tay, Louis, Adam W Meade, and Mengyang Cao. “An Overview and Practical Guide to IRT Measurement Equivalence Analysis.” Organizational Research Methods 18, no. 1 (January 1, 2015): 3–46. https://doi.org/10.1177/1094428114553062.

Welzel, Christian, Lennart Brunkert, Stefan Kruse, and Ronald F. Inglehart. “Non-Invariance? An Overstated Problem With Misconceived Causes.” Sociological Methods & Research, March 24, 2021, 0049124121995521. https://doi.org/10.1177/0049124121995521.

---

## [Decision Letter · Decision Letter 1]

10 Nov 2021

The role of education, religiosity and development on support for violent practices among Muslims in thirty-five countries

PONE-D-21-14799R1

Dear Dr. Gullickson,

We’re pleased to inform you that your manuscript has been judged scientifically suitable for publication and will be formally accepted for publication once it meets all outstanding technical requirements.

Kind regards,

Bryan L. Sykes, Ph.D.

Academic Editor

PLOS ONE

Additional Editor Comments (optional):

Reviewers' comments:

Reviewer's Responses to Questions

**Comments to the Author**

1. If the authors have adequately addressed your comments raised in a previous round of review and you feel that this manuscript is now acceptable for publication, you may indicate that here to bypass the “Comments to the Author” section, enter your conflict of interest statement in the “Confidential to Editor” section, and submit your "Accept" recommendation.

Reviewer #1: All comments have been addressed

2. Is the manuscript technically sound, and do the data support the conclusions?

Reviewer #1: Yes

3. Has the statistical analysis been performed appropriately and rigorously? 

Reviewer #1: Yes

4. Have the authors made all data underlying the findings in their manuscript fully available?

Reviewer #1: Yes

5. Is the manuscript presented in an intelligible fashion and written in standard English?

Reviewer #1: Yes

6. Review Comments to the Author

Reviewer #1: Dear Authors,

Thanks for the revision. The revised draft looks appropriate for the publication.

Kind regards

7. PLOS authors have the option to publish the peer review history of their article (what does this mean?). If published, this will include your full peer review and any attached files.

Reviewer #1: **Yes: **Dr Rojan Afrouz

---

## [Editor Report · Acceptance letter]

12 Nov 2021

PONE-D-21-14799R1 

The role of education, religiosity and development on support for violent practices among Muslims in thirty-five countries 

Dear Dr. Gullickson:

I'm pleased to inform you that your manuscript has been deemed suitable for publication in PLOS ONE. Congratulations! Your manuscript is now with our production department. 

Kind regards, 

on behalf of

Dr. Bryan L. Sykes 

Academic Editor

PLOS ONE